# Relationship between the extent of vascular injury and the evolution of surgically induced osteochondrosis lesions in a piglet model

Ferenc Tóth[1]*, Erick O. Buko[1,2], Alexandra R. Armstrong[1], Casey P. Johnson[1,2]

1 Department of Veterinary Clinical Sciences, College of Veterinary Medicine, University of Minnesota, St. Paul, Minnesota, United States of America, 2 Center for Magnetic Resonance Research, University of Minnesota, Minneapolis, Minnesota, United States of America

* ftoth@umn.edu

**Data Availability Statement:** All relevant data are within the manuscript and its Supporting Information files.

## Abstract

Ostechondritis dissecans (OCD) is an orthopaedic disease characterized by formation of osteochondral defects in developing joints. Epiphyseal cartilage necrosis (osteochondrosis [OC]) caused by focal failure of vascular supply is the known precursor lesion of OCD, but it remains to be established how the severity of vascular failure drives lesion healing or progression. In the current study we have implemented a novel piglet model of induced osteochondrosis of the lateral trochlear ridge of the femur to determine the role that the extent of ischemia plays in the development and progression of OC/OCD lesions. Ten 4-week-old Yorkshire piglets underwent surgical interruption of the vascular supply to the entirety (n = 4 pigs) or the distal half (n = 6 pigs) of the lateral trochlear ridge of the femur. At 2, 6, and 12 weeks postoperatively, distal femora were evaluated by magnetic resonance imaging (MRI) to determine the fate of induced OC lesions. At 12 weeks, piglets were euthanized, and the surgical sites were examined histologically. After complete devascularization, lesion size increased between the 6- and 12-week MRI by an average of 24.8 mm$^2$ (95% CI: [-2.2, 51.7]; p = 0.071). During the same period, lesion size decreased by an average of 7.6 mm$^2$ (95% CI: [-24.5, 19.4]; p = 0.83) in piglets receiving partial devascularization. At 12 weeks, average ± SD lesion size was larger (p<0.001) in piglets undergoing complete (73.5 ± 17.6 mm$^2$) vs. partial (16.5 ± 9.8 mm$^2$) devascularization. Our study demonstrates how the degree of vascular interruption determines lesion size and likelihood of healing in a large animal model of trochlear OC.

## Introduction

*Osteochondritis dissecans* (OCD) is a developmental joint disorder that affects both young animals and children, particularly those engaged in athletic activities. It is characterized by formation of osteochondral flaps and/or fragments within joints and has a high propensity to progress to early onset osteoarthritis [1–5]. Clinically-apparent OCD is usually seen in children between 6 and 19 years of age [1], but precursor lesions, termed *osteochondrosis* (OC), have been identified in the distal femur as early as 2 years of age [6]. While the majority of

**Funding:** This study was funded by grants from the NIH/NIAMS (R56 AR078209) and NIH/OD (K01OD034070). The sponsors had no role in the study design, collection, analysis, and interpretation of data; in the writing of the manuscript; and in the decision to submit the manuscript for publication.

**Competing interests:** The authors have declared that no competing interests exist.

precursor lesions are known to heal across species, a subset progress to clinically apparent disease [3, 7, 8]. Unfortunately, pathophysiologic processes determining the fate of OC lesions between resolution and clinical progression are yet to be elucidated, hampering patient care, as illustrated by the absence of evidence-based guidelines for the management of juvenile patients with OC/OCD [9].

The paucity of available information on the pathophysiology of OC/OCD in children, along with the lack of evidence-based treatments for the disease, can largely be explained by the difficulties associated with performing invasive studies in young human patients. In fact, most of the recent insights gained into the pathogenesis and diagnosis of OC/OCD have been inspired by studies conducted using naturally-occurring or surgically-induced animal models [8, 10–15]. Histological studies performed at OC/OCD predilection sites in joint explants obtained from asymptomatic piglets and foals have shown that discrete areas of epiphyseal cartilage necrosis (termed *OC-latens*), caused by focal failure of the vascular supply, are the clinically-silent precursor lesions of OCD [6, 16–18]. Delayed conversion of OC-*latens* lesions into bone presents as a focal failure of endochondral ossification, creating a radiographically apparent lesion known as OC-*manifesta*. Progression to clinically apparent OCD occurs when articular cartilage overlying these precursor lesions is unable to resist loading, leading to its collapse and fragmentation. Factors purported to contribute to lesion progression include biomechanical trauma (as suggested by the higher incidence of OCD among subjects participating in athletic activities [2, 19]), genetic factors [20], and lesion size [8].

Proof-of-principle studies conducted in horses [21] and pigs [8] along with experiments performed in goats [12, 13] have confirmed the role vascular failure plays in the pathogenesis of OC by inducing *OC-latens* and *OC-manifesta* lesions through surgical interruption of the perichondrial blood supply to the epiphyseal growth cartilage of the developing stifle (knee) joints. Unfortunately, none of these studies examined how the extent of vascular injury determines the size and clinical course of the developing lesions, leaving a critical gap in knowledge. This shortcoming is likely explained by the fact that most of these experiments targeted the femoral condyles for lesion induction, where visualization of the perichondrial blood supply, and thus its incremental interruption, is difficult. Conversely, the developing femoral trochlea with its comparatively simple surgical access and a well-defined, visible perichondral vasculature [21] represents an optimal location for graded devascularization.

In the study reported here, we have implemented a novel piglet model of surgically induced osteochondrosis of the lateral trochlear ridge of the femur to determine how the extent of vascular injury affects the development and progression of OC/OCD lesions. We hypothesized that surgical interruption of the vascular supply limited to the distal 50% of the lateral trochlear ridge will result in the formation of small *OC-latens* lesions which will undergo spontaneous resolution, whereas devascularization of the entirety of the lateral trochlear ridge will lead to the development of large *OC-latens* lesions that will progress to extensive *OC-manifesta* and, occasionally, to clinically apparent OCD.

## Materials and methods

### Design

Domestic Yorkshire piglets (n = 10) aged 4 weeks were enrolled in this study. The study protocol was approved by the University of Minnesota Institutional Animal Care and Use Committee (protocol #: 2203–39910). After a 3-day acclimation period, piglets underwent surgical interruption of the vascular supply to the entirety (n = 4 pigs) or the distal half (n = 6 pigs) of the lateral trochlear ridge of the distal femur in a randomly selected pelvic limb. The contralateral limbs served as unoperated controls. At 2, 6, and 12 weeks postoperatively, piglets received

*in vivo* MRIs to monitor development and progression or resolution of induced OC lesions. At the conclusion of the 12-week MRI, piglets were euthanized and their stifle joints were harvested for histologic processing.

## Surgery

Piglets were premedicated with a combination of Telazol (4 mg/Kg), xylazine (2 mg/Kg) and buprenorphine (0.02 mg/Kg) administered intramuscularly and were orotracheally intubated. General anesthesia was maintained by inhalation of isoflurane vaporized in oxygen. After routine preparation for aseptic surgery, a randomly selected stifle joint was approached using an 8 cm long medial parapatellar skin incision. The femoropatellar joint was entered using a lateral parapatellar incision and the patella was luxated medially to expose the femoral trochlea. Using a #15 blade, the vascular supply to the entirety (n = 4 pigs) or the distal half (n = 6 pigs) of the epiphyseal cartilage of the lateral trochlear ridge was interrupted by resecting a corresponding segment of the perichondrium from the abaxial aspect of the lateral trochlear ridge containing the nutrient vessels (Fig 1). Hemostasis was provided by pressing a sterile 4×4 gauze over the exposed bone, and the joint was lavaged with sterile saline solution. At the conclusion of the procedure, the incision was closed in layers and piglets were allowed to recover from general anesthesia.

## MRI

Anesthetized piglets underwent *in vivo* MRI at 2, 6, and 12 weeks postoperatively. Piglets were positioned in dorsal recumbency in a 3T MRI scanner (MAGNETOM Prisma; Siemens Healthcare). An 18-channel ultraflex receiver coil was placed over the bilateral stifles for signal reception (with the exception that a 4-channel flex coil was used for the 2-week postoperative MRI studies of piglets 1 and 2). Each stifle was imaged individually as part of the same imaging session using the same protocol. Imaging sequences included: (i) quantitative T2 relaxation

| Intact vasculature | Complete devascularization | Partial devascularization |
|---|---|---|
| 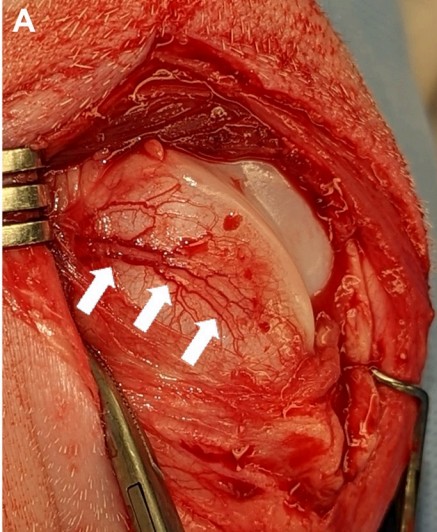 | 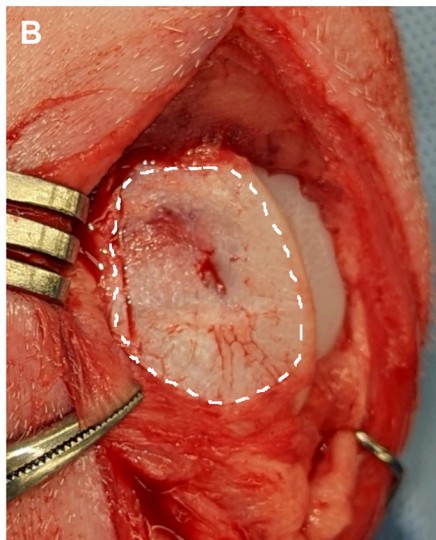 | 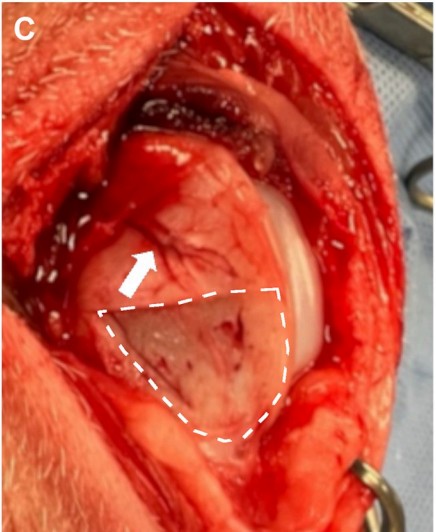 |

**Fig 1.** Intraoperative images showing: (A) the intact perichondrial vasculature (white arrows) of the lateral trochlear ridge; (B) complete devascularization of the lateral trochlear ridge (area marked with dashed line) after resection of perichondrium and the nutrient vessels contained within; and (C) partial devascularization of the lateral trochlear ridge (area marked by dashed line) and the intact proximal portion of the nutrient vessel (white arrow).

**Table 1. MRI scan parameters for one stifle.**

|  | MSME T2 Map | 3D DESS | Subtracted 3D CE-MRI |
|---|---|---|---|
| Field of view (mm$^3$) | 128×128×50 | 128×128×48 | 128×128×64 |
| Sampling matrix | 384×384×25 | 384×384×160 | 320×320×160 |
| Resolution (mm$^3$) | 0.3×0.3×2.0 | 0.3×0.3×0.3 | 0.4×0.4×0.4 |
| TR / TEs (ms) | 4000 / [11.5, 23.0, 34.5, 46.0, 57.5, 69.0, 80.5, 92.0] | 22.9 / 7.5 | 7.9 / 3.5 |
| Flip angle (°) | 90/180 | 25 | 25 |
| Bandwidth (Hz/px) | 250 | 128 | 250 |
| Fat settings | Fat saturation | Water excitation | - |
| Scan time (min) | 16 | 16 | 5.5 (×2 scans) |

Scan parameters are for the 2- and 6-week post-operative time points (6- and 10-week-old piglets). To accommodate the larger size of the piglets at the 12-week post-operative time point (16-week-old piglets), the field-of-view for the T2 map was increased to 140×140×50 mm$^3$ (giving 0.4×0.4×2.0 mm$^3$ resolution) and the field-of-view for the 3D DESS was increased to 140×140×64 mm$^3$ (giving 0.4×0.4×0.4 mm$^3$ resolution); all other scan parameters were fixed. The subtracted CE-MRI sequence was only acquired for the 2-week post-operative time point.

time mapping using a multi-slice multi-echo (MSME) spin echo sequence to identify *OC-latens* lesions in the epiphyseal cartilage; (ii) high-resolution 3D DESS to identify *OC-manifesta* lesions; and, following these sequences, (iii) subtracted contrast-enhanced MRI (CE-MRI) using a 3D GRE sequence acquired both before and one minute after intravenous administration of 0.2 mmol/kg gadoteridol contrast agent (ProHance; Bracco) to assess the extent of ischemia in the epiphyseal cartilage at the 2-week post-operative time point. Imaging parameters are shown in Table 1.

## Histology

Distal femoral specimens were fixed in 10% neutral buffered formalin for 72h, then the femoral trochleae were removed *in toto* from the parent bone and immersed in 10% ethylenediaminetetraacetic acid for decalcification. Decalcified specimens were serially sectioned in the sagittal plane into three to four 2.0 mm thick slabs that spanned the total width of the trochlea including the trochlear ridge and any grossly apparent lesion. Individual slabs were processed into paraffin blocks for histological evaluation. At least two 5-μm-thick sections were collected from the surface of each slab (n = 6 to 8 sections/trochlea) and stained with hematoxylin & eosin (H&E). Histological sections were assessed by a blinded, board-certified veterinary pathologist (ARA) with experience in musculoskeletal pathology. OC-*latens* lesions were defined as areas of chondronecrosis associated with necrotic vascular profiles that were confined to the epiphyseal cartilage [22]. OC-*manifesta* lesions were defined as areas of chondronecrosis in the epiphyseal cartilage that were accompanied by a delay in endochondral ossification [22]. Lesions were classified as OCD if they had the characteristics of OC-*manifesta* lesions accompanied by the presence of an osteochondral flap or fragment involving the articular surface.

## Data analysis

T2 relaxation time maps were generated offline using MATLAB (version 2023b; Mathworks) by fitting the echo time images to a mono-exponential signal decay model. Prior to fitting, the echo time images were denoised using TNORDIC to improve image quality [23] and the first echo time image was removed to reduce the influence of stimulated echoes. Subtracted CE-MRI images were generated by subtracting the pre-contrast 3D GRE scan from the identical post-contrast scan. CE-MRI studies were visually assessed to ensure potential motion did not result in misalignment between the pre- and post-contrast MR images.

For each trochlea, the MRI slice exhibiting the largest lesion in T2 relaxation time maps, as corroborated by the 3D DESS images, was determined. The lesion area on the slice was segmented and measured using ITK-SNAP (version 3.8.0; www.itksnap.org) [24]. Specifically, lesion segmentations were performed independently by two experienced investigators (FT and EOB), and one investigator repeated the segmentation, such that three lesion area measurements were made. The average of these three results was then reported for each trochlea.

The age and bodyweight of piglets at the time of the 12-week MRI study was compared between treatment groups using a two-tailed, unpaired t test. Lesion sizes were compared between and within treatment groups at 6 and 12 weeks postoperatively using a linear mixed effects model, with week, treatment, and their interaction as fixed effects, and animal as a random effect. Means and standard deviations are reported for each combination, along with pairwise comparisons for both treatments within week and weeks within treatment, with all p-values and confidence intervals adjusted using the Bonferroni-Sidak method. Given the small sample size and the multiple comparisons of interest, we chose to follow the guidelines in the 2019 American Statistician editorial [25] and not declare a predefined level of significance, but instead choose to "focus on the effect size and the uncertainty [to] help the reader understand the findings".

## Results

Mean ± SD age and weight of piglets at the time of surgery were 30.8 ± 3.3 days and 6.0 ± 1.1 kg. All surgical procedures were successfully completed without any complications. Qualitative evaluation of CE-MRI findings obtained 2 weeks postoperatively were consistent with localized ischemia of the lateral trochlear epiphyseal cartilage in 4/4 and 4/6 piglets undergoing complete and partial devascularization, respectively. Data obtained from these piglets with confirmed complete (piglets # 1–4) and partial ischemia (piglets # 5–8) were used to compare lesion sizes between treatment groups using the 6- and 12-week MRI findings. At the time of the 12-week MRI evaluation, the mean ± SD age of piglets receiving complete vs. partial devascularization was nearly identical at 115.8 ± 1.0 vs. 114.2 ± 3.8 days (p = 0.442). At the same timepoint, piglets appeared heavier in the complete devascularization group with a mean ± SD bodyweight of 57.3 ± 14.0 kg vs. 46.5 ± 9.7 kg for piglets undergoing partial devascularization; however, this difference was not statistically significant (p = 0.1858).

At 2 weeks postoperatively, qualitative analysis of CE-MRI findings and T2 maps were consistent with the presence of larger lesions in pigs undergoing complete vs. partial devascularization, but indistinct borders of the lesions did not permit quantitative comparisons. By 6 weeks after surgery, lesion borders became better defined, enabling consistent measurement of induced *OC-latens* lesions in the T2 maps (Table 2, Figs 2–5); mean ± SD lesion area in pigs undergoing complete devascularization was 48.8 ± 6.7 mm$^2$, consistent with an observed average difference of 24.7 mm$^2$ (95% CI: [-0.3, 49.8]; p = 0.053) when compared to partial devascularization (24.0 ± 11.4 mm$^2$). During the MRI examination conducted 12 weeks postoperatively, mean ± SD lesion area measurements obtained from T2 maps were greater in size in piglets receiving complete devascularization (73.5 ± 17.6 mm$^2$) compared to their measurements obtained at 6 weeks, by an average of 24.8 mm$^2$ (95% CI: [-2.2, 51.7]; p = 0.071), and also greater in size than measurements obtained in piglets 12 week after partial devascularization of the lateral trochleae, by an average of 57.1 mm$^2$ (95% CI: [23.1, 82.1]; p<0.001). Conversely, in pigs undergoing partial devascularization, mean ± SD lesion size at 12 weeks after surgery remained similar (n = 2) or moderately decreased (n = 2) relative to that measured at 6 weeks, with an average change of -7.6 mm$^2$ (95% CI: [-24.5, 19.4]; p = 0.83).

**Table 2. Osteochondrosis lesion sizes.**

| Pig # | Devascularization | | Lesion size [mm$^2$] | | Histology |
|:---:|:---:|:---:|:---:|:---:|:---:|
| | | | 6 weeks post-op | 12 weeks post-op | |
| 1 | complete | | 42.6 | 63.0 | OCM |
| 2 | complete | | 56.6 | 99.3 | OCM |
| 3 | complete | | 43.8 | 70.2 | OCM |
| 4 | complete | | 52.3 | 61.6 | OCM |
| | | mean ± SD | 48.8 ± 6.7 | 73.5 ± 17.6 | |
| 5 | partial | | 18.3 | 23.0 | OCM |
| 6 | partial | | 25.6 | 26.7 | OCM |
| 7 | partial | | 39.3 | 7.2 | OCM |
| 8 | partial | | 13.1 | 9.0 | OCM |
| | | mean ± SD | 24.0 ± 11.4 | 16.5 ± 9.8 | |
| 9 | partial | | NSF | NSF | OCM |
| 10 | partial | | NSF | NSF | NSF |

Measurements were taken in T2 maps at 6 and 12 weeks after partial or complete devascularization of the lateral trochlear ridge of the femur in ten piglets. Lesion types observed on histology at 12 weeks postoperatively are indicated in the last column (OCM: OC-*manifesta*; NSF: no significant finding).

Histological findings obtained 12 weeks after surgical induction of complete or partial ischemia of the lateral trochlear ridge were consistent with the MRI findings in all but 1 piglet (piglet #9). Large *OC-manifesta* lesions involving most of the lateral trochlear ridge were present in piglets receiving complete devascularization. In 4 piglets undergoing partial devascularization, *OC-manifesta* lesions were small, nearly completely surrounded by subchondral bone, including one piglet where MRI did not identify any lesions (piglet #9). One of the two remaining piglets receiving partial devascularization (piglet #6) had a medium sized *OC-*

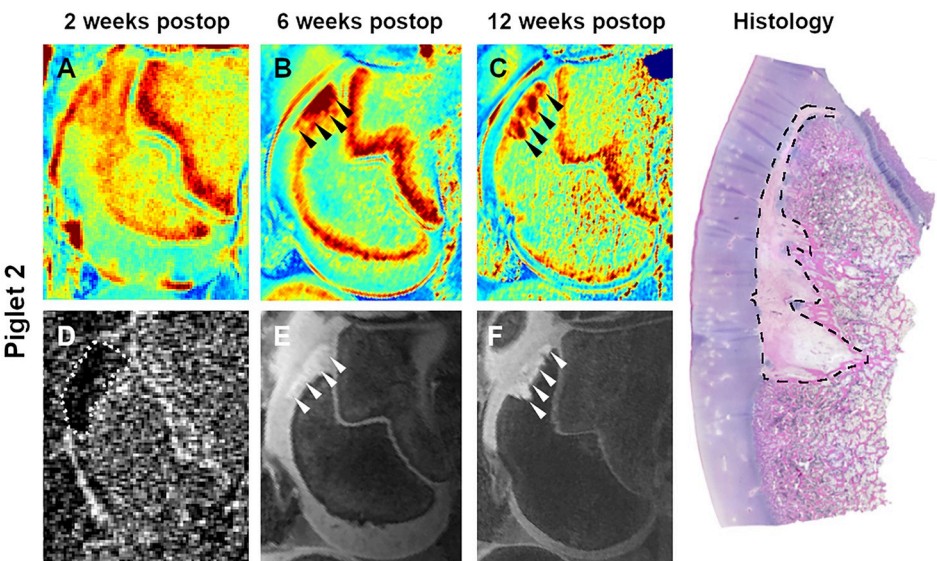

**Fig 2.** T2 cartilage maps (A-C), subtracted CE-MRI image (D), and 3D DESS images (E and F) obtained 2, 6, and 12 weeks after complete devascularization of the lateral trochlear ridge in piglet 2. Dotted line in the subtracted CE-MRI image (D) marks the extent of hypoperfusion. Arrowheads identify a large OC-latens (B and E) and a subsequent OC-manifesta (C and F) lesion. Photomicrograph shows the histologic appearance of the extensive OC-manifesta lesion (dashed line) at 12 weeks postoperatively.

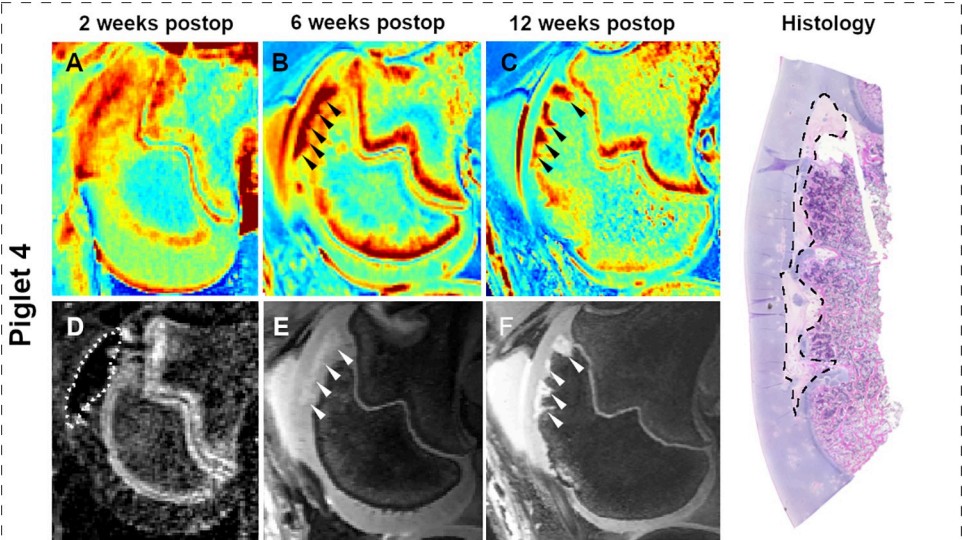

**Fig 3.** T2 cartilage maps (A-C), subtracted CE-MRI image (D), and 3D DESS images (E and F) obtained 2, 6, and 12 weeks after complete devascularization of the lateral trochlear ridge in piglet 4. Dotted line in the subtracted CE-MRI image (D) marks the extent of hypoperfusion. Arrowheads identify a large OC-latens (B and E) and a subsequent OC-manifesta (C and F) lesion. Photomicrograph shows the histologic appearance of a large OC-manifesta lesion (dashed line) at 12 weeks postoperatively.

*manifesta* lesion involving a substantial portion of the epiphyseal cartilage consistent with the MRI findings, whereas in the other piglet (piglet #10) no significant histologic or MRI findings were noted (**Fig 6**). In all *OC-manifesta* lesions, retention of necrotic cartilage matrix within

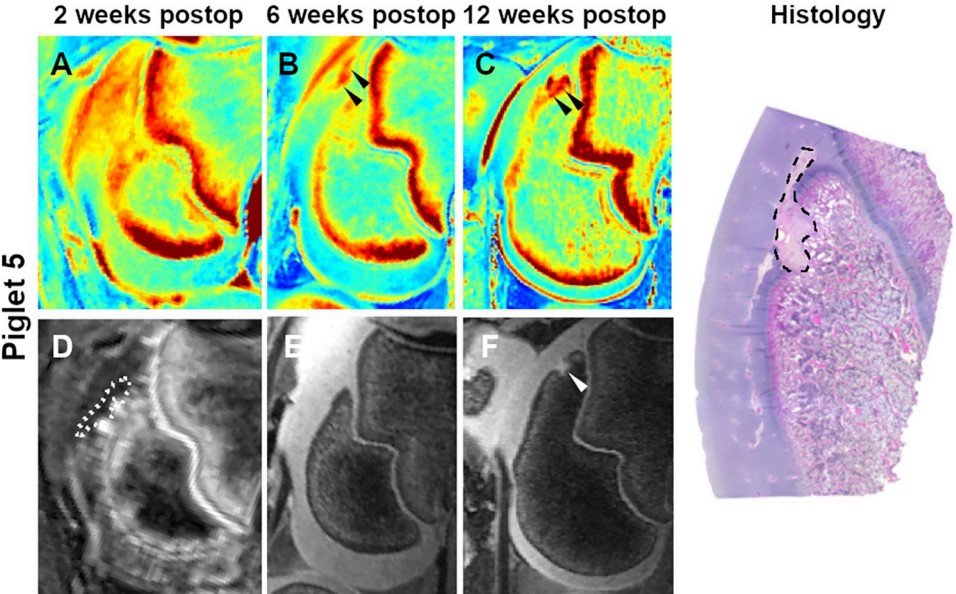

**Fig 4.** T2 cartilage maps (A-C), subtracted CE-MRI image (D), and 3D DESS images (E and F) obtained 2, 6, and 12 weeks after partial devascularization of the lateral trochlear ridge in piglet 5. Dotted line in the subtracted CE-MRI image (D) marks the extent of hypoperfusion. Arrowheads identify a small OC-latens (B) and a subsequent healing OC-manifesta (C and F) lesion. Photomicrograph shows the histologic appearance of the healing OC-manifesta lesion (dashed line) at 12 weeks postoperatively.

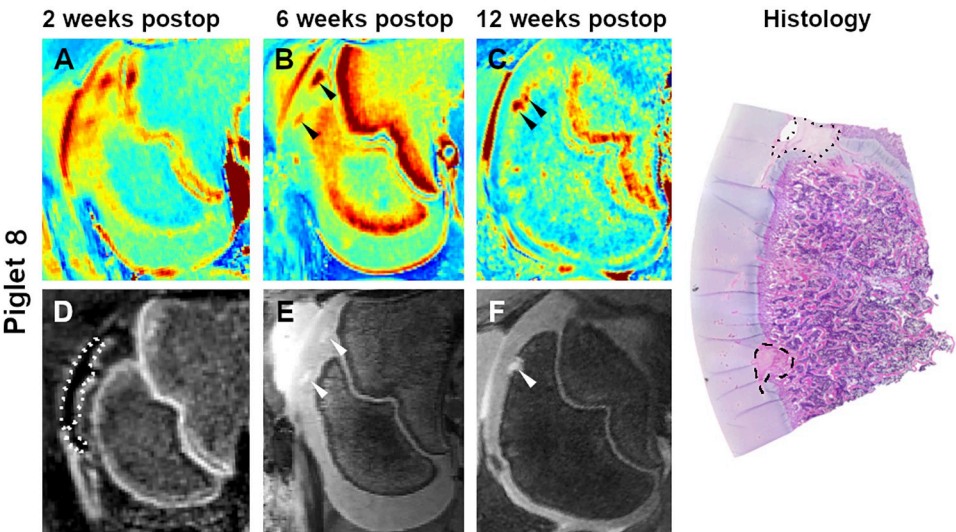

**Fig 5.** T2 cartilage maps (A-C), subtracted CE-MRI image (D), and 3D DESS images (E and F) obtained 2, 6, and 12 weeks after partial devascularization of the lateral trochlear ridge in piglet 8. Dotted line in the subtracted CE-MRI image (D) marks the extent of hypoperfusion. Arrowheads identify a small OC-latens (B) and a subsequent healing OC-manifesta (C and F) lesion. Photomicrograph shows the histologic appearance of the healing OC-manifesta lesion (dashed line) at 12 weeks postoperatively. A small OC lesion involving the growth plate (dotted line) is also apparent on histology but not observed on MRI.

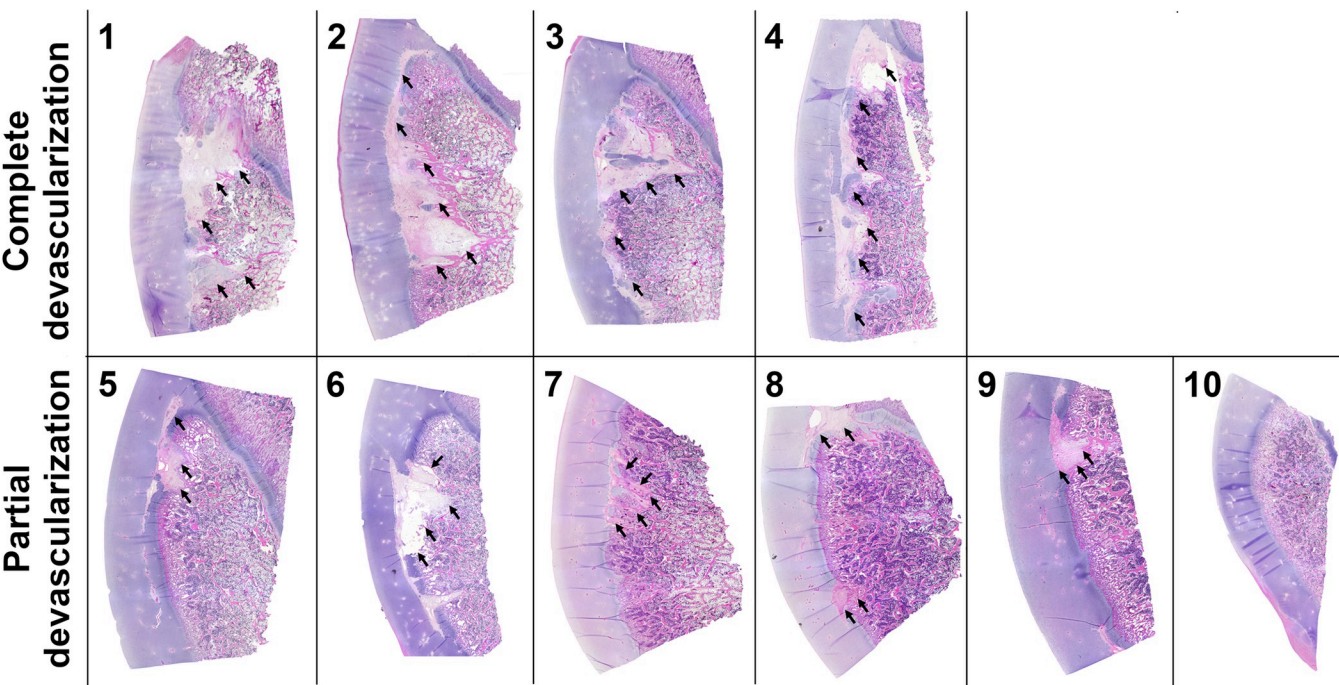

**Fig 6. Representative histological images of all 10 piglets obtained 12 weeks after they underwent complete or partial devascularization of the lateral trochlear ridge of the femur.** Complete devascularization in Piglets # 1, 2, 3 and 4 led to the formation of extensive *OC-manifesta* lesions involving most of the lateral trochlear ridge. After partial devascularization, Piglet # 6 developed a medium sized *OC-manifesta* lesion involving a substantial portion of the epiphyseal cartilage; Piglets # 5, 7, 8, and 9 had healing *OC-manifesta* lesions nearly completely surrounded by bone; and Piglet # 10 had no apparent lesion on histology. Lesions are marked with black arrows.

the epiphyseal cartilage overlying the area of delayed endochondral ossification was apparent. Adjacent to these affected areas, normal epiphyseal cartilage had few remaining intact vascular canals along with small numbers of chondrified and/or chondrifying vascular canals.

Neither MRI nor histology identified any lesions in the contralateral unoperated control limbs.

## Discussion

In the current study, we have successfully implemented a novel large animal model of OC through surgical interruption of the vascular supply to the epiphyseal cartilage of the lateral trochlear ridge of the femur using 4-week-old piglets. Our results obtained from this model demonstrate that increasing the extent of vascular injury is associated with the development of larger *OC-latens* lesions, which over time progress to extensive *OC-manifesta* lesions. Conversely, lesions induced by a more limited vascular injury result in smaller *OC-latens* and *OC-manifesta* lesions that are gradually incorporated in the subchondral bone by 12 weeks postoperatively as they undergo resolution.

The overwhelming majority of previous studies using an animal model to investigate the pathophysiology of osteochondrosis concentrated on the femoral condyles [8, 12, 13, 26, 27], the primary predilection site of OCD in children [1]. Studies conducted in goats using vascular interruption alone [12], or in combination with a controlled exercise regimen [13], were successful in inducing *OC-latens* lesions that progressed to small *OC-manifesta* lesions over time. Nevertheless, most of these induced lesions were comparatively small, and with a rare exception, underwent partial or complete resolution by 12 weeks postoperatively. A unique vascular architecture supplying the distal femoral epiphyseal cartilage in goats, starkly different from that seen in children [11], was suspected to drive the hastened healing process, and motivated a shift away from this species.

Follow-up investigations into the vascular supply of the epiphyseal growth cartilage at OCD predilection sites [8, 11, 28] identified (miniature) pigs as the ideal species to model OCD based on their shared distal femoral microvascular architecture with children [8, 11]. Notwithstanding the similarities in vascular anatomy, a subsequent study attempting to induce OCD lesions by interrupting the blood supply to the medial and lateral femoral condyles in juvenile miniature pigs was met with limited success, as all but one lesion were nearly or completely resolved by 10 weeks postoperatively. Furthermore, induced lesions were small, bringing their clinical importance into question, a likely consequence of a limited surgical access to the axial and abaxial vascular beds of the femoral condyles. These findings provided a strong impetus to target lesion induction to an alternative site in the stifle joint, which gained further support based on a previous proof-of-principle study that demonstrated successful induction of OCD-like lesions in the femoral trochlea in foals [21]. Indeed, perichondrial vessels supplying the developing femoral trochlea are visible on the abaxial aspect of the lateral trochlear ridge in pigs, lending themselves to relatively simple surgical interruption as demonstrated in the current study. Additionally, piglets also represent an economical alternative to foals, increasing the likelihood of adoption of the proposed model.

Importantly, a simple surgical access combined with the apparent visibility of perichondrial vasculature also allowed us to control the extent of induced ischemia. Considering the proximo-distal direction of blood flow, we were able to induce ischemia of the entirety or the distal half of the trochlea by interrupting the vessels either at the proximal extent or at the midpoint of the lateral trochlear ridge, respectively. This approach enabled us to investigate the role the extent of ischemia plays in the development, progression, and/or resolution of OC. Our findings are in line with the anecdotal evidence that small lesions are more likely to resolve,

whereas large lesions tend to persist or further increase in size resulting in extensive *OC-mani-festa* lesions, which is likely associated with an increased vulnerability to biomechanical trauma.

It has been purported that conversion of subclinical *OC-manifesta* lesions to clinically apparent OCD is promoted by exposure to biomechanical trauma, as suggested by an increased incidence of OCD in athletes [2, 3]. In our previous study, we attempted to evaluate the role of biomechanical trauma in the clinical progression of surgically induced OC lesions using a goat model, but differences in the vascular architecture and small lesion size interfered with our efforts [13]. Due to their large size and propensity to persist, *OC-manifesta* lesions of the lateral trochlear ridge induced in the current study represent an excellent opportunity to further explore the effects of biomechanical stress on the development of clinically apparent OCD lesions. For example, the superficial location of the lesion immediately beneath the patella will allow evaluation of the differences that compressive vs. shear forces (antero-poste-rior vs. latero-lateral impact) may have on the progression or resolution of *OC-manifesta* lesions.

Along with its many inherent advantages, the trochlear location of the induced lesions may also be perceived as a weakness of our study, given that this area is less frequently involved in naturally occurring OCD in children than the femoral condyles. Nevertheless, translation of findings obtained from the proposed model to human patients will not be limited to the less frequently seen trochlear lesions [1], but will also apply to femoral condylar OCD, due to the shared etiology and pathogenesis of OCD across its predilection sites [20]. Another limitation of the study is the insufficient follow-up period to allow complete healing and/or clinical pro-gression to occur. The absence of other predisposing factors (e.g., biomechanical trauma) likely contributed to our failure to observe clinical progression of lesions induced by complete devascularization, even though their size and MRI and histologic appearance are highly sug-gestive that at least a subset may have eventually formed osteochondral flaps and/or fragments as typically occurs in clinical disease. Lastly, while the low number of piglets enrolled in the study prevented us from conclusively showing how lesion size changes over time within groups, it was sufficient to demonstrate that more extensive vascular injury will result in a larger precursor lesion by 12 weeks postoperatively.

Taken together, our study provides new evidence that the extent of vascular injury determines lesion size, which is in turn an important factor in determining the fate of induced *OC-latens* lesions, with small lesions showing signs of resolution over time and large lesions progressing to extensive *OC-manifesta*. Additionally, we also describe a novel piglet model of trochlear OC that will allow opening new lines of investigations into the pathophysiology of OC/OCD.

## Supporting information

**S1 Dataset.**
(XLSX)

## Acknowledgments

We are grateful to Drs. Cathy Carlson and Aaron Rendahl for their invaluable contribution to the study design, interpretation of histological findings and statistical analysis of our data. We also thank the members of the University of Minnesota College of Veterinary Medicine Clini-cal Investigation Center for their assistance with the surgical procedures as well as Paula Overn and Katalin Kovacs, PhD for their help with histological processing of harvested specimens.

## Author Contributions

**Conceptualization:** Ferenc Tóth.

**Formal analysis:** Ferenc Tóth, Erick O. Buko, Alexandra R. Armstrong, Casey P. Johnson.

**Funding acquisition:** Ferenc Tóth.

**Writing – original draft:** Ferenc Tóth.

**Writing – review & editing:** Erick O. Buko, Alexandra R. Armstrong, Casey P. Johnson.

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
