## [Decision Letter · Decision Letter 0]

25 Jun 2024

PONE-D-24-17211Relationship Between the Extent of Vascular Injury and the Evolution of Surgically Induced Osteochondrosis Lesions in a Piglet Model.PLOS ONE

Dear Dr. Toth,

Thank you for submitting your manuscript to PLOS ONE. After careful consideration, we feel that it has merit but does not fully meet PLOS ONE’s publication criteria as it currently stands. Therefore, we invite you to submit a revised version of the manuscript that comprehensively addresses the minor points raised during the review process. Please submit your revised manuscript by Aug 09 2024 11:59PM. If you will need more time than this to complete your revisions, please reply to this message or contact the journal office at plosone@plos.org. Please include the following items when submitting your revised manuscript:A rebuttal letter that responds to each point raised by the academic editor and reviewer(s). You should upload this letter as a separate file labeled 'Response to Reviewers'.A marked-up copy of your manuscript that highlights changes made to the original version. You should upload this as a separate file labeled 'Revised Manuscript with Track Changes'.An unmarked version of your revised paper without tracked changes. You should upload this as a separate file labeled 'Manuscript'.If applicable, we recommend that you deposit your laboratory protocols in protocols.io to enhance the reproducibility of your results. Protocols.io assigns your protocol its own identifier (DOI) so that it can be cited independently in the future. For instructions see: https://journals.plos.org/plosone/s/submission-guidelines#loc-laboratory-protocols. Additionally, PLOS ONE offers an option for publishing peer-reviewed Lab Protocol articles, which describe protocols hosted on protocols.io. Read more information on sharing protocols at https://plos.org/protocols?utm_medium=editorial-email&utm_source=authorletters&utm_campaign=protocols.

We look forward to receiving your revised manuscript.

Kind regards,

Andre van Wijnen

Academic Editor

PLOS ONE

Journal Requirements:

2. Thank you for stating the following in the Acknowledgments Section of your manuscript: "We are grateful to Drs. Cathy Carlson and Aaron Rendahl for their invaluable contribution to the study design, interpretation of histological findings and statistical analysis of our data. We also thank the members of the University of Minnesota College of Veterinary Medicine Clinical Investigation Center for their assistance with the surgical procedures as well as Paula Overn and Katalin Kovacs, PhD for their help with histological processing of harvested specimens. This study was funded by grants from the NIH/NIAMS (R56 AR078209) and NIH/OD (K01OD034070). The sponsors had no role in the study design, collection, analysis, and interpretation of data; in the writing of the manuscript; and in the decision to submit the manuscript for publication. The authors have no conflict of interest to declare."

Please remove any funding-related text from the manuscript and let us know how you would like to update your Funding Statement. Currently, your Funding Statement reads as follows: "This study was funded by grants from the NIH/NIAMS (R56 AR078209) and NIH/OD (K01OD034070). The sponsors had no role in the study design, collection, analysis, and interpretation of data; in the writing of the manuscript; and in the decision to submit the manuscript for publication."

3. We note that your Data Availability Statement is currently as follows: "All relevant data are within the manuscript and its Supporting Information files."

Reviewers' comments:

Reviewer's Responses to Questions

**Comments to the Author**

1. Is the manuscript technically sound, and do the data support the conclusions?

Reviewer #1: Yes

Reviewer #2: Yes

2. Has the statistical analysis been performed appropriately and rigorously? 

Reviewer #1: Yes

Reviewer #2: I Don't Know

3. Have the authors made all data underlying the findings in their manuscript fully available?

Reviewer #1: Yes

Reviewer #2: Yes

4. Is the manuscript presented in an intelligible fashion and written in standard English?

Reviewer #1: Yes

Reviewer #2: Yes

5. Review Comments to the Author

Reviewer #1: This is a good researched and well written manuscript describing a novel piglet model of very rare but debilitating disease - osteochondritis dissecans. Authors explore the role of extent of vascular injury in the pathophysiology of OCD. I only have a few minor suggestions for improvement and clarification that should be addressed before publishing this article in PLOS ONE. See my comments below:

1) Intro – ln 43: please flip the order to “both young animals and children, particularly…”. I believe authors refer to children rather ten animals engaged in athletic activities.

2) Methods / MRI – ln 127: It seems that two different flex coils were used to accommodate imaging of larger pigs at 12 weeks. I would recommend stating this fact in the Methods section.

3) Methods / MRI – ln 133: Please provide more info on CE-MRI. What was the contrast dose administered? Were CE images acquired after the acquisition of DESS and T2 images? What was the time delay before contrast injection and start of post-CE MRI?

4) Methods / Data analysis – ln 166: Were pre- and post-contrast MR images registered before calculating the subtraction image? This was in vivo scan. Although the subtracted CE-MR images were analyzed only qualitatively, it would be helpful to describe measures taken to account for potential motion between the pre- and post-CE MRI.

5) Results - ln 212: change “the same” to more fitting “similar”.

6) Results - ln 255: authors probably meant “(piglet #9)” instead of “(piglet #6)”.

7) References #23, #27 and #28 seems incomplete; please check.

8) Figures 2E,F; 3F; 4F; 5F – please change contrast in these images to better depict cartilage and lesions. Figure 3E looks so much better than 2E.

9) Figure 6 - Please add arrows pointing to OCD lesions.

Reviewer #2: This is an interesting and well performed study, which demonstrates that a total vascular interruption to the trochlea ridge of femur will cause larger cartilage lesions than a partial vascular insult. The surgical procedure with a control of the extent of induced ischemia to the growth cartilage of the femoral trochlear is ingenious.

The hypothesis is clear and it is unlikely to expect a different outcome. The physiological laws that an insult to a larger vascular bed should cause a larger infarct seems obvious. But as stated in the discussion (line 308) it is “anecdotal evidence”.

The data shows that the lesions after a partial devascularization is indeed smaller, but also decrease over time which is the most important finding. It would have been even more interesting to follow the piglets longer, with MR images, to see if the lesions eventually would disappear completely in this group and if the larger lesion would have developed into OCD.

Below see a few suggestions/concerns on the manuscript

Specific comments to the authors:

The abstract is clearly written and describes the study. I think a more precise description of the breed of “novel piglet model” could be added already in the abstract.

Suggestion: Line 29-30….Ten 4-week-old Yorkshire piglets underwent…..

The introduction describes the relevant known knowledge of the disease, including accurate references, about osteochondrosis. The author’s hypothesis is clear. They want to demonstrate that partial vascular interruption will cause small OC lesions and that they will decrease in size over time. An interruption of the entire vascularization to the lateral trochlea will cause larger OC lesions that increase in size and develop into OC-manifesta.

Material & Methods.

The description of the material and methods are clear, with a few clarifications suggested.

Line 149: … sectioned into serial 2,0 mm thick slabs that spanned the total width of the trochlea… How many slabs did this result in from each trochlea?

Line 150: Out of these, two 5-µm-thick slabs were processed for histology from each surface of the 2-mm slab. How many histologic sections were evaluated from each trochlea?

The OCM on histology were graded in size (result-section). Small, moderate? Large? Measured?

Data analysis: I am not an expert in this field, hence these analysis and results must be evaluated by another referee. The sample size is small, but the data appear to be consistent and Table 2 clearly shows difference in size between the groups, which is not surprising.

Results:

The description of results are mostly clear. I only have a few questions/concerns.

The weight and exact age of the pigs at the time of surgery is stated in line 189.

How much did the different groups weigh at euthanasia and were all 12,0 weeks?

Is it possible to define the histologic OCM size more precisely?

Small, medium and large sized OCM? Perhaps add a description of the different sizes more clearly in the legends to figure 6, then the reader will understand the different sizes more clearly. The following description could be used in fig. 6: Line 251: large OCM = “most of the lateral trochlear ridge. Line 253: Small= nearly completely surrounded by bone. Line 255: Medium OCM = a substantial portion of the cartilage.

The exact size of the MRI lesions are more precise, since they are measured.

What did the growth cartilage, overlying the OCM look like? Was there resolved vascularization, total disappearance of vessels with chondrification or still a large OCLatens, with necrotic cartilage? Did the control side (not operated) still show vascularization at 12-weeks or a complete lack of vessels or chondrificaiton?

Was the overlying non-vascularized articular cartilage affected at all in the operated side?

Discussion:

The surgical procedure with a control of the extent of induced ischemia to the growth cartilage of the femoral trochlear is ingenious and opens up for more research into OC.

Lines 319-322: The discussion about the different forces (compression and shear) involved in the progression of OCM to OCD is of great importance, and can hopefully be addressed using the presented piglet model.

The authors do discuss the limitations of the study, low number of pigs, the lack of the follow up to healing or development of an OCD and the location of the femoral trochlea instead of the condyles, more predisposed in humans. However in equines, the femoral trochlea is predisposed to OCD with subsequent clinical lameness, hence I think it is of value to also explore this location.

References:

It does include the relevant references

6. PLOS authors have the option to publish the peer review history of their article (what does this mean?). If published, this will include your full peer review and any attached files.

Reviewer #1: No

Reviewer #2: No

---

## [Author Response · Author response to Decision Letter 0]

8 Jul 2024

Response to reviewers

The authors would like to thank the reviewers for their thorough work. Their comments were instrumental in helping the authors to improve on the original submission in the revised manuscript.

Reviewer 1

“This is a good researched and well written manuscript describing a novel piglet model of very rare but debilitating disease - osteochondritis dissecans. Authors explore the role of extent of vascular injury in the pathophysiology of OCD. I only have a few minor suggestions for improvement and clarification that should be addressed before publishing this article in PLOS ONE. See my comments below”

R1.1: Reviewer’s comment: L42 - Please flip the order to “both young animals and children, particularly…”. I believe authors refer to children rather then animals engaged in athletic activities. 

Authors’ response: Thank you for your comment, the text has been changed to reflect your suggestion.

Changes made in manuscript: L42-43: “Osteochondritis dissecans (OCD) is a developmental joint disorder that affects both young animals and children, particularly…” 

R1.2: Reviewer’s comment: L127 - It seems that two different flex coils were used to accommodate imaging of larger pigs at 12 weeks. I would recommend stating this fact in the Methods section. 

Authors’ response: All but two MRI studies were acquired using an 18-channel ultraflex receiver coil. The first two piglets had their 2-week post op MRI study conducted using a 4-channel flex coil, as we had yet to acquire the 18-channel ultraflex one. The 18-channel ultraflex coil was used for the rest of the studies for its improved signal-to-noise ratio. 

Changes made in manuscript: L127-129: “An 18-channel ultraflex receiver coil was placed over the bilateral stifles for signal reception (with the exception that a 4-channel flex coil was used for the 2-week postoperative MRI studies of piglets 1 and 2).

R1.3: Reviewer’s comment: L133 - Please provide more info on CE-MRI. What was the contrast dose administered? Were CE images acquired after the acquisition of DESS and T2 images? What was the time delay before contrast injection and start of post-CE MRI? 

Authors’ response: Thank you for your comment. CE images were acquired after the DESS and T2 images. There was a one-minute delay between the contrast injection (0.2 mmol/kg gadoteridol [ProHance; Bracco]) and the start of the post-CE MRI.

Changes made in manuscript: L133-138: “and, following these sequences, (iii) subtracted contrast-enhanced MRI (CE-MRI) using a 3D GRE sequence acquired both before and one minute after intravenous administration of 0.2 mmol/kg gadoteridol contrast agent (ProHance; Bracco) to assess the extent of ischemia…”

R1.4: Reviewer’s comment: L166 - Were pre- and post-contrast MR images registered before calculating the subtraction image? This was in vivo scan. Although the subtracted CE-MR images were analyzed only qualitatively, it would be helpful to describe measures taken to account for potential motion between the pre- and post-CE MRI. 

Authors’ response: The subtracted images were visually assessed for any misalignments that could have been caused by potential motion between the pre- and post-CE MRI. No such motion was observed for any of the acquisitions.

Changes made in manuscript: L172-173: “CE-MRI studies were visually assessed to ensure potential motion did not result in misalignment between the pre- and post-contrast MR images.” 

R1.5: Reviewer’s comment: L212 - change “the same” to more fitting “similar” 

Authors’ response: Thank you for your comment, the text has been changed to reflect your suggestion. 

Changes made in manuscript: L224: “…after surgery remained similar (n = 2) or moderately decreased (n = 2) relative to…”

R1.6: Reviewer’s comment: L255 - authors probably meant “(piglet #9)” instead of “(piglet #6). 

Authors’ response: Thank you for the comment. We believe that the piglet numbers are correctly referenced here. Of 6 piglets that underwent partial devascularization, 4 piglets had very small OC-manifesta lesions (including piglet #9 whose lesion was inapparent on MRI). No lesions were found either histologically or on MRI in piglet #10. Conversely, piglet #6 had a medium sized OCM lesion. We believe that these findings can also be observed in Figure 6. Please let us know if we are overlooking something here.

Changes made in manuscript: L273: No changes were made here. 

R1.7: Reviewer’s comment: References - #23, #27 and #28 seems incomplete; please check

Authors’ response: Thank you for your suggestion. The above references have been double checked. Reference #28 was in print at the time of the submission of the original manuscript. It is now published, and the reference was updated with all the available new information. Reference #27 was published in an online journal; therefore, page numbers are not available. Nevertheless, this reference was supplemented with a doi number to ease its identification. Reference #23 is a preprint document, so there is limited information available but a doi number has also been added to the original reference. 

Changes made in manuscript: references:

“23. Moeller S, Buko EO, Parvaze SP, Dowdle L, Ugurbil K, Johnson CP, et al. Locally low-rank denoising in transform domains. bioRxiv. 2023. doi: 10.1101/2023.11.21.568193.

27. Pfeifer CG, Kinsella SD, Milby AH, Fisher MB, Belkin NS, Mauck RL, et al. Development of a Large Animal Model of Osteochondritis Dissecans of the Knee: A Pilot Study. Orthop J Sports Med. 2015;3(2):2325967115570019. doi: 10.1177/2325967115570019.

28. Tóth F, Nissi MJ, Armstrong AR, Buko EO, Johnson CP. Epiphyseal cartilage vascular architecture at the distal humeral osteochondritis dissecans predilection site in juvenile pigs. J Orthop Res. 2024; 42(3):737-744.”

R1.8: Reviewer’s comment: Figures 2E,F; 3F; 4F; 5F – please change contrast in these images to better depict cartilage and lesions. Figure 3E looks so much better than 2E. 

Authors’ response: Thank you for your comment, we have attempted to adjust the brightness and contrast in Figures 2E,F; 3F; 4F; 5F to better match the native contrast of Figure 3E

Changes made in manuscript: Figures 2E,F; 3F; 4F; 5F: Adjustments were made as requested to the contrast and brightness levels in Figures 2E,F; 3F; 4F; 5F 

R1.9: Reviewer’s comment: Figure 6 - Please add arrows pointing to OCD lesions. 

Authors’ response: OC-manifesta lesions are marked with arrows in the revised figure.

Changes made in manuscript: Figure 6: Arrows marking OC-manifesta lesions have been added to figure 6. 

Reviewer 2

“This is an interesting and well performed study, which demonstrates that a total vascular interruption to the trochlea ridge of femur will cause larger cartilage lesions than a partial vascular insult. The surgical procedure with a control of the extent of induced ischemia to the growth cartilage of the femoral trochlear is ingenious.

The hypothesis is clear, and it is unlikely to expect a different outcome. The physiological laws that an insult to a larger vascular bed should cause a larger infarct seems obvious. But as stated in the discussion (line 308) it is “anecdotal evidence”.

The data shows that the lesions after a partial devascularization is indeed smaller, but also decrease over time which is the most important finding. It would have been even more interesting to follow the piglets longer, with MR images, to see if the lesions eventually would disappear completely in this group and if the larger lesion would have developed into OCD.”

R2.1: Reviewer’s comment: The abstract is clearly written and describes the study. I think a more precise description of the breed of “novel piglet model” could be added already in the abstract.

Suggestion: Line 29-30…Ten 4-week-old Yorkshire piglets underwent 

Authors’ response: Thank you for your comment, the text has been changed to reflect your suggestion.

Changes made in manuscript: L27-28: “Ten 4-week-old Yorkshire piglets…” 

R2.2: Reviewer’s comment: L149 - sectioned into serial 2,0 mm thick slabs that spanned the total width of the trochlea… How many slabs did this result in from each trochlea? 

Authors’ response: The number of slabs slightly varied by the size of the trochlea, but typically ranged from 3-4 2mm slabs, and always included the trochlear ridge and all grossly apparent lesions.

Changes made in manuscript: L153-156: “Decalcified specimens were serially sectioned in the sagittal plane into three to four 2.0 mm thick slabs that spanned the total width of the trochlea including the trochlear ridge and any grossly apparent lesion.” 

R2.3: Reviewer’s comment: L150 - Out of these, two 5-µm-thick slabs were processed for histology from each surface of the 2-mm slab. How many histologic sections were evaluated from each trochlea?

The OCM on histology were graded in size (result-section). Small, moderate? Large? Measured? 

Authors’ response: The number of slabs per harvested stifle ranged from three to four and from each of these slabs two 5-um thick sections were processed and assessed histologically; therefore, 6-8 sections were examined from each trochlea. OCM lesion sizes were assessed qualitatively as small, medium, or large; they were not measured on histology. Please see comment R2.5 for details. (Quantitative assessment of lesion size was based on the MRI findings, because it allowed precise evaluation of each individual slice, thus identification of the slice with the largest apparent lesion area in each pig.) 

Changes made in manuscript: L157-158: “At least two 5-μm-thick sections were collected from the surface of each slab (n=6 to 8 sections/ trochlea) and stained with hematoxylin & eosin (H&E).” 

R2.4: Reviewer’s comment: L189 - The weight and exact age of the pigs at the time of surgery is stated in line 189. How much did the different groups weigh at euthanasia and were all 12,0 weeks? 

Authors’ response: Thank you for your comment, the requested information has been included in the revised manuscript.

Changes made in manuscript: L204-209: “At the time of the 12-week MRI evaluation, the mean ± SD age of piglets receiving complete vs. partial devascularization was nearly identical at 115.8 ± 1.0 vs. 114.2 ± 3.8 days (p = 0.442). At the same timepoint, piglets appeared heavier in the complete devascularization group with a mean ± SD bodyweight of 57.3 ± 14.0 kg vs. 46.5 ± 9.7 kg for piglets undergoing partial devascularization; however, this difference was not statistically significant either (p = 0.1858).” and L184-185: “The age and bodyweight of piglets at the time of the 12-week MRI study was compared between treatment groups using a two-tailed, unpaired t-test.” 

R2.5: Reviewer’s comment: L251-255 - Small, medium and large sized OCM? Perhaps add a description of the different sizes more clearly in the legends to figure 6, then the reader will understand the different sizes more clearly. The following description could be used in fig. 6: Line 251: large OCM = “most of the lateral trochlear ridge. Line 253: Small= nearly completely surrounded by bone. Line 255: Medium OCM = a substantial portion of the cartilage. 

Authors’ response: The legend for Figure 6 has been amended by the suggested descriptors of the lesions. 

Changes made in manuscript: Figure 6: “Fig 6. Representative histological images of all 10 piglets obtained 12 weeks after they underwent complete or partial devascularization of the lateral trochlear ridge of the femur. Complete devascularization in Piglets #1, 2, 3, and 4 led to the formation of extensive OC-manifesta lesions involving most of the lateral trochlear ridge. After partial devascularization, Piglet #6 developed a medium-sized OC-manifesta lesion involving a substantial portion of the epiphyseal cartilage; Piglets #5, 7, 8, and 9 had healing OC-manifesta lesions nearly completely surrounded by bone; and Piglet #10 had no apparent lesion on histology.” 

R2.6: Reviewer’s comment: Results general - What did the growth cartilage, overlying the OCM look like? Was there resolved vascularization, total disappearance of vessels with chondrification or still a large OCL, with necrotic cartilage? Did the control side (not operated) still show vascularization at 12-weeks or a complete lack of vessels or chondrificaiton? Was the overlying non-vascularized articular cartilage affected at all in the operated side? 

Authors’ response: OC-manifesta lesions typically included necrotic cartilage extending into the epiphyseal cartilage overlying the area of delayed endochondral ossification. Within the epiphyseal cartilage superficial to the OC-manifesta lesions (i.e., between the articular surface and the necrotic epiphyseal cartilage), there were few persistent small vascular canals along with small numbers of vascular canals that were either already chondrified or beginning to chondrify. On the control side, persistent vascular canals along with scattered chondrified vascular canals were both present in the epiphyseal growth cartilage at the time of sampling (16 weeks of age; 12 weeks post-operative). There were no significant changes to the overlying articular cartilage at the operated sites.

Changes made in manuscript: L313-317: “In all OC-manifesta lesions, retention of necrotic cartilage matrix within the epiphyseal cartilage overlying the area of delayed endochondral ossification was apparent. Adjacent to these affected areas, normal epiphyseal cartilage had few remaining intact vascular canals along with small numbers of chondrified and/or chondrifying vascular canals.”

Minor formatting changes were also made in the manuscript to meet the requirements for publication in PLOS ONE.

---

## [Decision Letter · Decision Letter 1]

29 Jul 2024

Relationship Between the Extent of Vascular Injury and the Evolution of Surgically Induced Osteochondrosis Lesions in a Piglet Model.

PONE-D-24-17211R1

Dear Dr. Toth,

We’re pleased to inform you that your manuscript has been judged scientifically suitable for publication and will be formally accepted for publication once it meets all outstanding technical requirements.

Kind regards,

Andre van Wijnen

Academic Editor

PLOS ONE

Additional Editor Comments (optional):

Reviewers' comments:

Reviewer's Responses to Questions

**Comments to the Author**

1. If the authors have adequately addressed your comments raised in a previous round of review and you feel that this manuscript is now acceptable for publication, you may indicate that here to bypass the “Comments to the Author” section, enter your conflict of interest statement in the “Confidential to Editor” section, and submit your "Accept" recommendation.

Reviewer #1: All comments have been addressed

Reviewer #2: All comments have been addressed

2. Is the manuscript technically sound, and do the data support the conclusions?

Reviewer #1: Yes

Reviewer #2: Yes

3. Has the statistical analysis been performed appropriately and rigorously? 

Reviewer #1: Yes

Reviewer #2: Yes

4. Have the authors made all data underlying the findings in their manuscript fully available?

Reviewer #1: Yes

Reviewer #2: Yes

5. Is the manuscript presented in an intelligible fashion and written in standard English?

Reviewer #1: Yes

Reviewer #2: Yes

6. Review Comments to the Author

Reviewer #1: (No Response)

Reviewer #2: The authors have responded to my questions and added explanations in the text as well as in the figure legends.

I am not an expert in statistics, but to my knowledge it appears sound

7. PLOS authors have the option to publish the peer review history of their article (what does this mean?). If published, this will include your full peer review and any attached files.

Reviewer #1: No

Reviewer #2: No

---

## [Editor Report · Acceptance letter]

1 Aug 2024

PONE-D-24-17211R1 

PLOS ONE

Dear Dr. Tóth, 

I'm pleased to inform you that your manuscript has been deemed suitable for publication in PLOS ONE. Congratulations! Your manuscript is now being handed over to our production team.

Kind regards, 

on behalf of

Dr. Andre van Wijnen 

Academic Editor

PLOS ONE